# Towards Interpretable Reinforcement Learning Using Attention Augmented Agents

**Alex Mott\*    Daniel Zoran\*    Mike Chrzanowski**

**Daan Wierstra    Danilo J. Rezende**
DeepMind
London, UK
`{alexmott,danielzoran,chrzanowski,wierstra,danilor}@google.com`

## Abstract

Inspired by recent work in attention models for image captioning and question answering, we present a soft attention model for the reinforcement learning domain. This model uses a soft, top-down attention mechanism to create a bottleneck in the agent, forcing it to focus on task-relevant information by sequentially querying its view of the environment. The output of the attention mechanism allows direct observation of the information used by the agent to select its actions, enabling easier interpretation of this model than of traditional models. We analyze different strategies that the agents learn and show that a handful of strategies arise repeatedly across different games. We also show that the model learns to query separately about space and content ("where" vs. "what"). We demonstrate that an agent using this mechanism can achieve performance competitive with state-of-the-art models on ATARI tasks while still being interpretable.

## 1   Introduction

Traditional RL agents and image classifiers rely on some combination of convolutional and fully connected components to gradually process input information and arrive at a set of policy or class logits. This sort of architecture is very effective, but does not lend itself to easy understanding of how decisions are taken, what information is used and why mistakes are made. Previous efforts to visualize deep RL agents [1, 2, 3] focus on generating saliency maps to understand the magnitude of policy changes as a function of a perturbation of the input. This can uncover some of the "attended" regions, but may be difficult to interpret. It also can't reveal certain types of behavior, such as when the agent makes decisions based on components *absent* from a frame. The model we propose here provides a more direct interpretation by making the attention an explicit bottleneck in the system.

In this work we apply a soft, top-down, spatial attention mechanism to visual information in a reinforcement learning setting. The model enables us to build agents that actively select important, task-relevant information from visual inputs by sequentially querying and receiving compressed, query-dependent summaries to generate appropriate outputs. While doing this, the model generates attention maps which can uncover some of the underlying decision process used to solve the task. By observing and analyzing the resulting attention maps we can understand *how* the system solves a task. We observe that the attention focuses on the key components of each level: tracking the region ahead of the player, focusing on enemies and important moving objects. We find that the agent reacts in a consistent manner even when encountering new, unseen configurations of the environment. We also observe that the agent is able to localize its attention based on both the *content* in the frame as well as absolute *spatial positions*. Finally we find that building the attention into the agent yields more

---

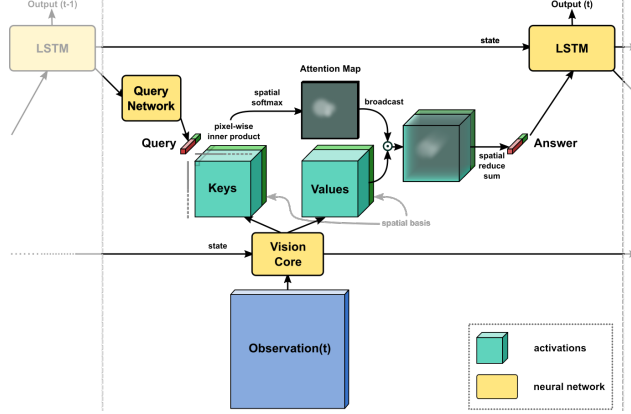

Figure 1: An outline of our proposed model. Observations pass through a recurrent vision core network, producing a "keys" and a "values" tensor, to both of which we concatenate a spatial basis tensor (see text for details). A recurrent network at the top sends its state from the previous time-step into a query network which produces a set of query vectors (only one is shown here for brevity). We calculate the inner product between each query vector and each location in the keys tensor, then take the spatial softmax to produce an attention map for the query. The attention map is broadcast along the channel dimension, point-wise multiplied with the values tensor and the result is then summed across space to produce an answer vector. This answer is sent to the top LSTM as input to produce the output and next state of the LSTM. We omit some extra inputs to the LSTM (such as previously taken action and reward) for clarity, see text for full details.

informative visualizations and gives more assurance that attended regions are indeed the cause for the agent's actions than other methods for analyzing saliency.

## 2   Model

Our model, outlined in Figure 1, **queries** a large input tensor through an attention mechanism and uses the returned **answer** (a low dimensional vector summary of the input based on the query) to produce its output. We refer to this full query-answer system as an **attention head**. Our system can implement multiple attention heads by producing multiple queries and receiving multiple answers.

An observation $\mathbf{X} \in \mathbb{R}^{H \times W \times C}$ at time $t$ (here an RGB frame of height $H$ and width $W$) is passed through a "vision core". The vision core is a multi-layer convolutional network $\mathrm{vis}_\theta$ followed by a recurrent layer with state $\boldsymbol{s}_{\mathrm{vis}}(t)$ such as a ConvLSTM [4], which produces an output tensor $\mathbf{O}_{\mathrm{vis}} \in \mathbb{R}^{h \times w \times c}$:

$$\mathbf{O}_{\mathrm{vis}}, \boldsymbol{s}_{\mathrm{vis}}(t) = \mathrm{vis}_\theta(\mathbf{X}(t), \boldsymbol{s}_{\mathrm{vis}}(t-1)) \tag{1}$$

The vision core output is then split along the channel dimension into two tensors: the "Keys" tensor $\mathbf{K} \in \mathbb{R}^{h \times w \times c_K}$ and the "Values" tensor $\mathbf{V} \in \mathbb{R}^{h \times w \times c_v}$, with $c = c_V + c_K$. To the keys and values tensors we concatenate a spatial basis — a fixed tensor $\mathbf{S} \in \mathbb{R}^{h \times w \times c_S}$ which encodes spatial locations (see below for details).

An LSTM with parameters $\phi$ produces $N$ queries, one for each attention head. The LSTM sends its state $\boldsymbol{s}_{\mathrm{LSTM}}$ from the previous time step $t-1$ into a "Query Network". The query network $Q_\psi$ is a multi-layer perceptron (MLP) with parameters $\psi$ whose output is reshaped into $N$ query vectors $\boldsymbol{q}^n$ of size $c_K + c_S$ such that they match the channel dimension of $K \in \mathbb{R}^{h \times w \times c_K + c_S}$, which is the concatenation along the channel dimension of $\mathbf{K}$ and $\mathbf{S}$.

$$\boldsymbol{q}^1 ... \boldsymbol{q}^N = Q_\psi(\boldsymbol{s}_{\mathrm{LSTM}}(t-1)) \tag{2}$$

Similar to [5], we take the inner product between each query vector $\boldsymbol{q}^n$ and all spatial locations in the keys tensor $K$ to form the $n$-th attention logits map $\tilde{\boldsymbol{A}}^n \in \mathbb{R}^{h \times w}$:

$$\tilde{A}^n_{i,j} = \sum_l q^n_l K_{i,j,l} \tag{3}$$

We then take the spatial softmax to form the final normalized attention map $\boldsymbol{A}^n$:

$$A^n_{i,j} = \frac{\exp(\tilde{A}^n_{i,j})}{\sum_{i',j'} \exp(\tilde{A}^n_{i',j'})} \quad (4)$$

Each attention map $\boldsymbol{A}^n$ is broadcast along the channel dimension of the values tensor $\mathsf{V} \in \mathbb{R}^{h \times w \times C_v + C_S}$ (the concatenation along the channel dimension of $\mathsf{V}$ and $\mathsf{S}$), point-wise multiplied with it and then summed across space to produce the $n$-th answer vector $\boldsymbol{a}^n \in \mathbb{R}^{1 \times 1 \times c_v + c_s}$:

$$a^n_c = \sum_{i,j} A^n_{i,j} \mathsf{V}_{i,j,c} \quad (5)$$

Finally, the $N$ answer vectors $\boldsymbol{a}^n$, and the $N$ query vectors form the input to the LSTM to produce the next LSTM state $\boldsymbol{s}_{\text{LSTM}}(t)$ and output $\boldsymbol{o}(t)$ for this time step:

$$\boldsymbol{o}(t), \boldsymbol{s}_{\text{LSTM}}(t) = \text{LSTM}_\phi(a^1, ..., a^n, q^1, ..., q^n, \boldsymbol{s}_{\text{RNN}}(t-1)) \quad (6)$$

The exact details for each of the networks, outputs and states are given in Section 4 and the Appendix.

It is important to emphasize several points about the proposed model. First, the model is fully differentiable due to the use of soft-attention and can be trained using back-propagation. Second, the query vectors are a function of the LSTM state alone and not the observation — this allows for a "top-down" mechanism where the RNN can actively query the input for task-relevant information rather than having to filter out large amounts of information. Third, the spatial sum (Equation 5) is a severe spatial bottleneck, which forces the system to make the attention maps in such a way that information is not "blurred" out during summation.

The summation of the values tensor of shape $h \times w \times c_v$ to an answer of shape $1 \times 1 \times c_v$ is invariant to permutation of spatial position, which creates the need for the spatial basis. The only way the LSTM can know and reason about spatial positions is if the spatial information is encoded in the values of the tensors $\mathsf{K}$ and $\mathsf{V}$ [1]. We postulate that the query and answer structure can have different "modes" — the system can ask "where" something is by sending out a query with zeros in the spatial channels of the query and non-zeros in the channels corresponding to the keys (which are input dependent). It can then read the answer from the spatial channels, localizing the object of interest. Conversely it can ask "what is in this particular location" by zeroing out the content channels of the query and putting information on the spatial channels, reading the content channels of the answer and ignoring the spatial channels. This is not a dichotomy as the two can be mixed (e.g. "find enemies in the top left corner"), but it does point to an interesting "what" and "where" separation, which we discuss in Section 4.6.

## 2.1 The spatial basis

The spatial basis $\mathsf{S} \in \mathbb{R}^{h \times w \times c_S}$ is structured such that the channels at each location $i, j$ encode information about the spatial position. Concatenating this information into the values of the tensor allows some spatial information to be maintained after the spatial summation (Equation 5) removes the structural information. Following [5] and [6] we use a Fourier basis type of representation. Each channel $(u, v)$ of $\mathsf{S}$ is an outer product of two Fourier basis vectors. We use both odd and even basis functions with several frequencies. For example, with two even functions one channel of $\mathsf{S}$ with spatial frequencies $u$ and $v$ would be:

$$\mathsf{S}_{i,j,(u,v)} = \cos(\pi u i / h) \cos(\pi v j / w) \quad (7)$$

where $u, v$ are the spatial frequencies in this channel, $i, j$ are spatial locations in the tensor and $h, w$ are correspondingly the height and width of the tensor. We produce all the outer products such that the number of channels in $\mathsf{S}$ is $(U + V)^2$ where $U$ and $V$ are the number of spatial frequencies we use for the even and odd components (4 for both throughout this work, so 64 channels in total).

## 3 Related work

There is a vast literature on recurrent attention models. They have been applied with some success to question-answering datasets [7, 8], text translation [5, 9], video classification and captioning [10, 11],

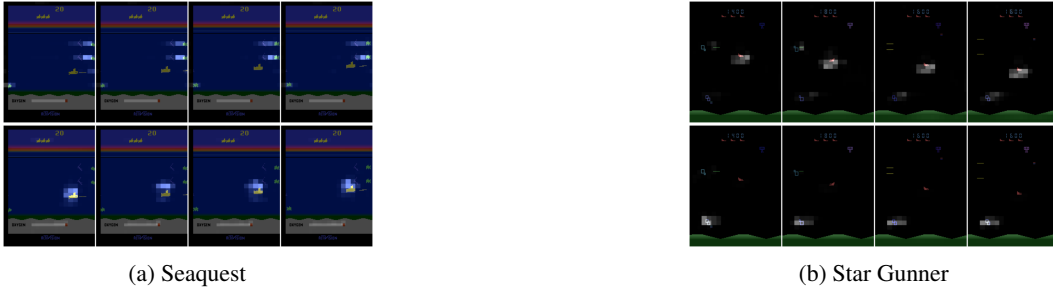

| (a) Seaquest | (b) Star Gunner |

Figure 2: Basic attention patterns. Bright areas are regions of high attention. Here we show 2 of the 4 heads used (one head in each row, time goes from left to right). The model learns to attend key sprites such as the player and different enemies. Best viewed on a computer monitor. See text for more details.

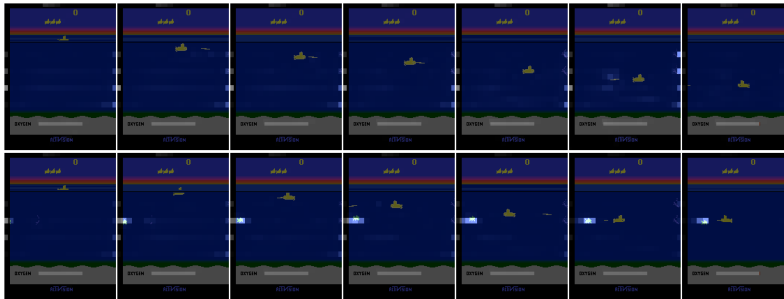

Figure 3: Reaction to novel states. A sequence of Seaquest frames as produced by the environment (top row) and with an additional fish injected into the image (bottom row). The fish is added coming from the left side of the screen at the beginning of the episode (which never occurs during training). The agent is able to react, attend to, and fire at the new object. Moreover, when the object is not destroyed by its first shot, it turns back to the object and fires again.

image classification and captioning [12, 13, 14, 15, 16, 17, 18, 19, 20], text classification [21, 22], generative models [6, 23, 24], object tracking [25], and reinforcement learning [26, 27, 28]. These attention mechanisms can be grouped by whether they use hard attention (e.g. [12, 20, 29]) or soft attention (e.g. [9]) and whether they explicitly parameterize an attention window (e.g. [30, 10]) or use a weighting mechanism (e.g. [5, 7]).

We use a soft key, query, and value type of attention similar to [5] and [6], but instead of doing "self"-attention where the queries come from the input (together with the keys and values) our queries come from a top-down source which does not directly depend on the input. This enables our system to be both state/context dependent and input dependent. Furthermore the output of the attention model is highly compressed and has no spatial structure (other than the one encoded using the spatial basis), unlike in "self" attention models where each pixel attends to every other pixel and the spatial structure is preserved.

Our model is most similar to the MAC model [31] and the Show Attend and Tell model [19], with several important adaptations to make it suitable to reinforcement learning. MAC was built to solve CLEVR [32], while Show Attend and Tell was designed for image captioning; major parts of each are geared for the task they are trying to solve. MAC's "control" unit is built to expect a guiding question for the reasoning process, which does not exist in the RL case; our system needs to come up with its own queries to produce the required output. In Show Attend and Tell, a fixed image is used, so there is no need to process multiple objects at the same time. Furthermore, neither model needs to track motion nor store absolute spatial position, both of which are important attributes for an agent.

## 4   Analysis and Results

We use the Arcade Learning Environment [33] to train and test our agent on 57 different Atari games. The model uses a 3 layer convolutional neural network followed by a convolutional LSTM as the

---

All referenced videos can be found at `https://sites.google.com/view/s3ta`.

vision core. Another (fully connected) LSTM generates a policy $\pi$ and a baseline function $V^\pi$ as output; it takes as input the query and answer vectors, the previous reward and a one-hot encoding of the previous action. The query network is a three layer MLP, which takes as input the hidden state $h$ of the LSTM from the previous time step and produces 4 attention queries. See Appendix A.1 for a full specification of the network sizes.

We use the Importance Weighted Actor-Learner Architecture [34] training architecture to train our agents. We use an actor-critic setup and a VTRACE loss with an RMSProp optimizer (see learning parameters in Appendix A.1 for more details).

We compare against two models without attentional bottlenecks to benchmark performance, both using the deeper residual network described in [34]. In the Feedforward Baseline, the output of the ResNet is used to directly produce $\pi$ and $V^\pi$, while in the LSTM Baseline an LSTM with 256 hidden units is inserted on top of the ResNet. The LSTM also gets as input the previous action and previous reward. We find that our agent is competitive with these state-of-the-art baselines, see Table 1 for benchmark results and Appendix A.3 for learning curves and performance on individual levels. Our

Table 1: Human normalized scores for experts on ATARI.

| Model | Median | Mean |
|---|---|---|
| Feedforward Baseline | 284.5% | 1479.5% |
| LSTM Baseline | 45.0% | 1222.0% |
| Attention | **407.1%** | **1649.0%** |

main focus is on analyzing the attention maps produced by our agent as it solves these tasks. Though these maps do not necessarily tell the whole story of decision making, they do expose some of the strategies used by the model. In order to visualize the attention maps we show the original input frame and super-impose the attention map $A^n$ for each head on it using alpha blending. This means that the bright areas in all images are the ones which are attended to, darker areas are not.

We find the range of values to be such that areas which are not attended have weights very close to zero, meaning that little information is "blended" from these areas during the summation in Equation 5. We validate this in Section A.6.

## 4.1 Basic attention patterns

The most dominant pattern we observe is that the model learns to attend to task-relevant things in the scene. In most ATARI games that usually means that the player is one of the foci of attention, as well as enemies, power-ups and the score itself (which is an important factor in the calculating the value function). Figure 2 (best viewed on screen) shows several examples of these attention maps. We also recommend watching the videos posted online for additional visualizations.

## 4.2 Reaction to novel states

The Atari environment is often quite predictable: enemies appear at regular times and in regular configurations. It is, therefore, important to ensure that the model truly learns to attend the objects of interest and act upon the information, rather than memorize or react to certain patterns in the game. In other words, we want to see that what the agent attends to has a direct influence on the agent's actions, rather than just a correlation with them.

In order to test this we injected an enemy object (a fish in Seaquest) to the observation at an unexpected time and in an unexpected location. This was done just at the pixel level, not at the game engine level, so it's not a "real" game object. In Figure 3, we see that the agent is able to attend and react to the introduction of the new enemy in an appropriate way. The agent attends the fish, moves toward it, fires at it, and then turns away. When it observes that it did not destroy the fish (since we simply spliced the fish into the video feed, not into the engine), it turns back and fires again. This demonstrates that the agent can generalize to unseen configurations — actively using its attention, reacting to new information, and acting appropriately — rather than memorizing fixed patterns of behavior.

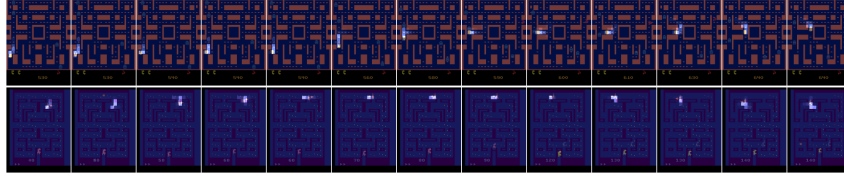

Figure 4: Forward planning/scanning. We observe that in games where there is a clear mapping between image space and world space and some planning is required, the model learns to scan through possible future trajectories for the player and chooses ones that are safe/rewarding. The images show two such examples from Ms Pacman and Alien. Note how the paths follow the map structure. See text for more details and videos. Bright areas are regions of high attention.

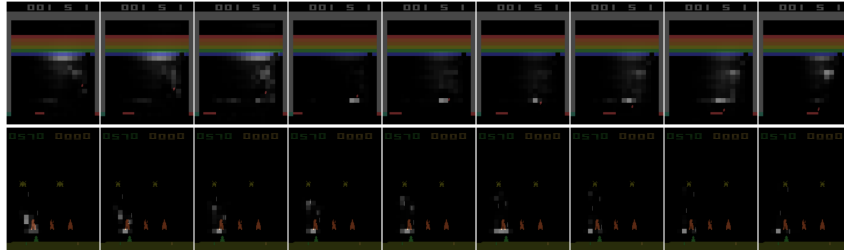

Figure 5: Trip Wires. We observe in games where there are moving balls or projectiles that the agent sets up tripwires to create an alert when the object crosses a specific point or line. The agent learns how much time it needs to react to the moving object and sets up a spot of attention sufficiently far from the player. In Breakout (top row), one can see a two level tripwire: initially the attention is spread out, but once the ball passes some critical point it sharpens to focus on a point along the trajectory, which is the point where the agent needs to move toward the ball. In Space Invaders (bottom row) we see the tripwire acting as a shield; when a projectile crosses this point the agent needs to move away from the bullet. Bright areas are regions of high attention.

### 4.3 Forward Planning/Scanning

In games where there is an element of forward planning and a direct mapping between image space and world space (such as 2D top-down view games) we observe that the model learns to scan through possible paths emanating from the player character and going through possible future trajectories. Figure 4 shows a examples of this in Ms Pacman and Alien — in the both games the model scans through possible paths, making sure there are no enemies or ghosts ahead. We observe that when it does see a ghost, another path is produced or executed in order to avoid it. Again we refer the reader to the videos for a better impression of the dynamics.

### 4.4 The role of top-down influence

To test the importance of the *top-down* nature of the queries, we train two additional agents with modified attention mechanisms that do not receive queries from the top-level RNN but are otherwise identical to our agent. The first agent uses the same attention mechanism except that the queries are a learnable bias tensor which does not depend on the LSTM state (this style of attention is similar to the one used in [26], although that model does not include many of the adaptations used here). The second agent does away with the query mechanism entirely and forms the weights for the attention by computing the L2 norm of each key (similar to a soft version of [29]). Both of these modifications turn the top-down attention into a bottom-up attention, where the vision network has total control over the attention weights.

We train these agents on 7 ATARI games for $2e9$ steps and compare the performance to the agent with top-down attention. We see significant drops in performance on 6 of the 7 games. On the remaining game, Seaquest, we see substantially improved performance; the positions of the enemies follow a very specific pattern, so there is little need for sequential decision making in that environment. On these games we see a median human normalized score of $541.1\%$ for the attention agent, $274.7\%$ for the fixed-query agent, and $274.5\%$ for the L2-Norm Key Agent. Mean scores are $975.5\%$, $615.2\%$ and $561.0\%$ respectively. See Appendix A.4 for more details.

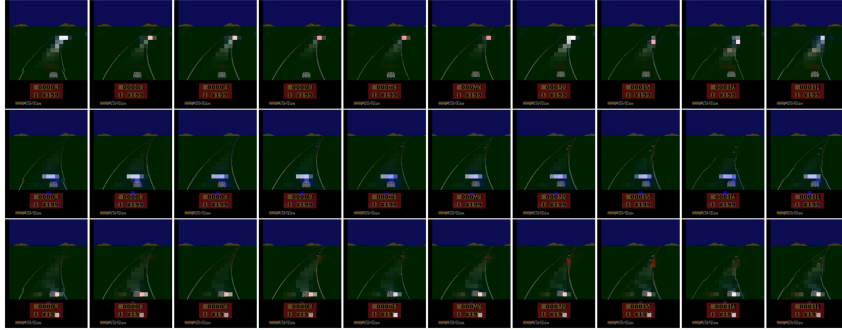

Figure 6: What/Where. This figures shows a sequence of 10 frames from Enduro (arranged left-to-right) along with the what-where visualization of each of the 3 of the 4 attention heads. (stacked vertically). The top row is the input frame at that timestep. Below we visualize the relative contribution of "what" vs. "where" in different attention heads: Red areas indicate the query has more weight in the "what" section, while blue indicates the mass is in the "where" part. White areas indicate that the query is evenly balanced between what and where. We notice that the first head here scans the horizon for upcoming cars and then starts tracking them (swithing from mixed to "what"). The second head is mostly a "where" query following the car for upcoming vehicles (a "trip-wire"). The last head here mostly tracks the player car and the score (mostly "what").

## 4.5 "Trip wires"

In many games we observe that the agent learns to place "trip-wires" at strategic points in space such that if a game object crosses them a specific action is taken. For example, in Space Invaders two such trip wires are following the player ship on both sides such that if a bullet crosses one of them the agent immediately evades them by moving towards the opposite direction. Another example is Breakout where we can see it working in two stages. First the attention is spread out around the general area of the ball, then focuses into a localized line. Once the ball crosses that line the agent moves towards the ball. Figure 5 shows examples of this behavior.

## 4.6 "What" vs. "Where"

As discussed in Section 2, each query has two components: one interacts with the keys tensor - which is a function of the input frame and vision core state - and the other interacts with the fixed spatial basis, which encodes locations in space. Since the output of these two parts is added together via an inner product prior to the softmax, we can analyze, for each query and attention map, which part of the query is more responsible for the the attention at each point; we can contrast the "what" from the "where". For example, a query may be trying to find ghosts or enemies in the scene, in which case the "what" component should dominate as these can reside in many different places. Alternatively, a query could ask about a specific location in the screen (e.g., if it plays a special role in a game), in which case we would expect the "where" part to dominate.

We visualize this by color coding the relative dominance of each part of the query. When a specific location is more influenced by the contents part, we will color the attention red, and when it is more influenced by the spatial part, we color it blue. Intermediate values will be white. More details can be found in Appendix A.5.

Figure 6 shows several such maps visualized in Enduro for different query heads. As can be seen, the system uses the two modes to make its decisions, some of the heads are content specific looking for opponent cars. Some are mixed, scanning the horizon for incoming cars and when found, tracking them, and some are location based queries, scanning the area right in front of the player for anything the crosses its path (a "trip-wire" which moves with the player). Examples of this mechanism in action can be seen in the videos online.

## 4.7 Comparison with other attention analysis methods

In order to demonstrate that the attention masks are an accurate representation of where the agent is looking in the image, we perform the saliency analysis presented in [1] on both the attention agent and the baseline feedforward agent. This analysis works by introducing a small, local Gaussian blur

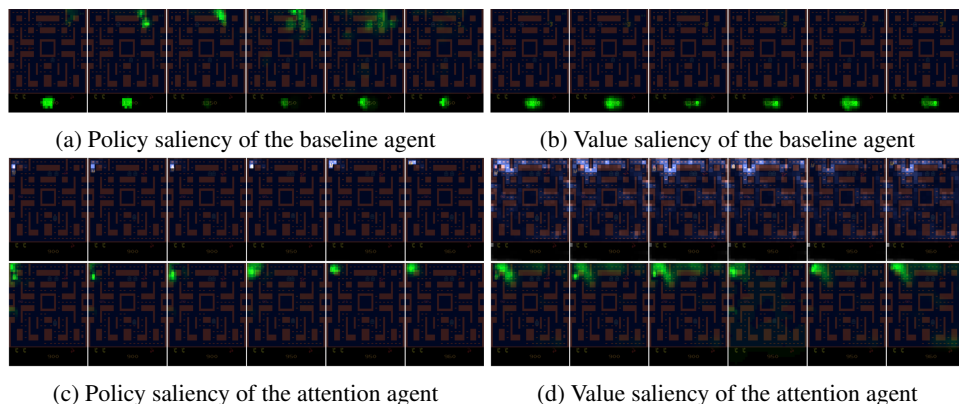

(a) Policy saliency of the baseline agent

(b) Value saliency of the baseline agent

(c) Policy saliency of the attention agent

(d) Value saliency of the attention agent

Figure 7: Saliency analysis. We visualize saliency (see text for details) in green. The saliency coming from the policy logits is mostly concerned with the area directly around Pacman in both the baseline (a) and attention (c) agents. The saliency in the attention agent is sharper than in the baseline and corresponds directly with one of the attention heads (highlighted in white), but the overall structure is similar. The saliency coming from the value function is very different in the baseline (b) and attention agent (d). In the baseline it is mostly concerned with the score. In the attention agent, the value saliency corresponds to the head that is looking further ahead (longer term planning/scanning behaviour), following different paths through the game map. This shows that indeed, this attention head contributes directly to the value estimate of the agent. See text for details and videos.

at a single point in the image and measuring the magnitude of the change in the policy. By measuring this at every pixel in the image, one can form a response map that shows how much the agent relies on the information at every spatial point to form its policy.

To produce these maps we run a trained agent for $> 200$ unperturbed frames on a level and then repeatedly input the final frame with perturbations at different locations. We form two saliency maps $S_\pi(i,j) = 0.5||\pi(\mathbf{X}'_{i,j}) - \pi(\mathbf{X})||^2$ and $S_{V^\pi}(i,j) = 0.5||V^\pi(\mathbf{X}'_{i,j}) - V^\pi(\mathbf{X})||^2$ where $\mathbf{X}'_{i,j}$ is the input frame blurred at point $(i,j)$, $\pi$ are the softmaxed policy logits and $V^\pi$ is the value function. An example of these saliency maps is shown in Figure 7. We see that the saliency map (in green) corresponds well with the attention map produced by the model and we see that the agent is sensitive to points in its planned trajectory, as we discussed in Section 4.3. Furthermore we see the heads specialize in their influence on the model — one clearly affects the policy more where the other affects the value function.

Comparing the attention agent to the baseline agent, we see that the attention agent is sensitive to more focused areas along the possible future trajectory. The baseline agent is more focused on the area immediately in front of the player (for the policy saliency) and on the score, while the attention agent focuses more specifically on the path the agent will follow (for the policy) and on possible future longer term paths (for the value).

## 5 Conclusion

We have applied an attention mechanism to an agent trained with reinforcement learning on the ATARI environment. The agent achieves performance competitive with state-of-the-art agents across a broad range of ATARI levels. The attention mechanism produces attention maps which can be used to visualize which parts of the input are attended. We have seen that the top-down nature of the attention provides a large performance gain compared to equivalent, bottom-up attention based mechanisms. We have seen that the agent is able to make use of a combination of "what" and "where" queries to select both regions and objects within the input depending on the task. We have also observed that the agents are able to learn to focus on key features of the inputs, look ahead along short trajectories, and place tripwires to trigger certain behaviors. Comparison of the attention maps to alternate methods for visualizing saliency shows that the attention allows more comprehensive analysis of the information the agent is using to inform its policy. We hope that model such as the one proposed here will help advance our understanding of agents and their underlying decision process.

## Footnotes

[1] The vision core might have some ability to produce information regarding absolute spatial positioning but, due to its convolutional structure, it is limited.

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
