[Supplementary Material]

## A Appendix

### A.1 Agent Description

Our agent takes in ATARI frames in RGB format ($210 \times 160 \times 3$) and processes them through a two layer ConvNet and a ConvLSTM, which produces an output of size $27 \times 20 \times 128$. We split this output along the channel dimension to produce keys of size $27 \times 20 \times 8$ and values of size $27 \times 20 \times 120$. To each of these we append the same spatial basis of size $27 \times 20 \times 64$. The query is produced by feeding the state of the LSTM after the previous time step to a three layer MLP. The final layer produces a vector with length 288, which is reshaped into a matrix of size $4 \times 72$ to represent the different attention heads. The queries, keys and values are processed by the mechanism described in Section 2 and produces answers. The queries, answers, previous action, and previous reward are fed into an answer processor, which is a 2 layer MLP. The output of the answer processor is the input to the policy core, which is an LSTM. The output of the policy core is processed through a one layer MLP and the output of that is processed by two different one layer MLPs to produce the policy logits and values estimate. All the sizes are summarizes in Table 2.

| Module | Type | Sizes |
|---|---|---|
| vision core<br>kernel size: $4 \times 4$, stride: 2, feature layers: 64 | CNN | [l]kernel size: $8 \times 8$, stride: 4, channels: 32 |
| vision RNN | ConvLSTM | kernel size: $3 \times 3$, channels: 128 |
| answer processor<br>hidden units: 256 | MLP | [l]hidden units: 512 |
| policy core | LSTM | hidden units: 256 |
| query network<br>hidden units: 128<br>hidden units: $72 \times 4$ | MLP | [l]hidden units: 256 |
| policy & value output | MLP | hidden units: 128 |

Table 2: The network sizes used in the attention agent

We use an RMSProp optimizer with $\epsilon = 0.01$, momentum of 0, and decay of 0.99. The learning rate is $2e - 4$. We use a VTRACE loss with a discount of 0.99 and an entropy cost of 0.01 (described in [33]); we unroll for 50 timesteps and batch 32 trajectories on the learner. We clip rewards to be in the range $[-1, 1]$, and clip gradients to be in the range $[-1280, 1280]$. Since the framerate of ATARI is high, we send the selected action to the environment 4 times without passing those frames to the agent in order to speedup learning. Parameters were chosen by performing a hyperparameter sweep over 6 levels (battle zone, boxing, enduro, ms pacman, seaquest, star gunner) and choosing the hyperparameter setting that performed the best on the most levels.

### A.2 Multi-Level Agents

We also train an agent on all ATARI levels simultaneously. These agents have distinct actors acting on different levels all feeding trajectories to the same learner. Following [33], we train the agent using population based training ([34]) with a population size of 16, where we evolve the learning rate, entropy cost, RMSProp $\epsilon$, and gradient clipping threshold. We initialize the values to those used for the single level experts, and let the agent train for $2e7$ frames before begining evolution. We use the mean capped human normalized score described in [33] to evaluate the relative fitness of each parameter set.

### A.3 Agent Performance

Figure 8 shows the training curves for the experts on 55 ATARI levels (the curves for Freeway and Venture are omitted since they are both constantly 0 for all agents). Table 1 shows the final human-normalized score achieved on each game by each agent in both the expert and multi-agent regime. As expected, the multi-level agent achieves lower scores on almost all levels than the experts.

Figure 8: Performance of individual experts on selected ATARI games. `Freeway` and `Venture` are omitted; no tested agent achieved a non-zero return on either game

## A.4 Top-Down versus Bottom-Up

Figure 9 shows the training curves for the Fixed Query Agent and the L2 Norm Keys agent. These agents are all trained on single levels for $2e9$ frames. We see that, in 6 of the 7 tested games, the agents without top-down attention perform significantly worse than the agent with top-down attention. Table 4 shows the final scores achieved by each agent on all 7 levels.

Figure 9: The role of top-down influence: Performance of individual experts on selected ATARI games.

## A.5  What-Where Analysis

To form the what-where maps shown in Section 4.6, we compute the relative contribution $C_{i,j}$ for a query $q$ from the content and spatial parts at each location is defined to be:

$$\text{what}_{i,j} = \sum_{h=1}^{C_k} q_h K_{i,j,h} \tag{8}$$

$$\text{where}_{i,j} = \sum_{h=1}^{C_s} q_{h+C_k} S_{i,j,h} \tag{9}$$

$$D_{i,j} = \begin{cases} -log(10) & \text{what}_{i,j} - \text{where}_{i,j} < -log(10) \\ \text{what}_{i,j} - \text{where}_{i,j} & |\text{what}_{i,j} - \text{where}_{i,j}| \leq log(10) \\ log(10) & \text{what}_{i,j} - \text{where}_{i,j} > log(10) \end{cases} \tag{10}$$

$$C_{i,j} = D_{i,j} A_{i,j} \tag{11}$$

where we interpolate between red, white and blue according to the values of $C$. The intuition is that, at blue (red) points the contribution from the spatial (content) portion to the total weights would be more than 10 times greater than the other portion. We truncate at $\pm 10$ because there are often very large differences in the logits, but after the softmax huge differences become irrelevant. We weight by the overall attention weight to focus the map only on channels that actually contribute to the overall weight map.

## A.6  Validity of attention maps

In order to demonstrate that the agent is mostly using the information contained in the regions of high attention, we re-run the trained agent with the attention modified to suppress areas with small attention weights. For this test, we substitute the attention weights $A_{i,h}^n$ in Equation 5 for

$$\tilde{\mathcal{A}}_{i,j}^n(t) = \begin{cases} A_{i,j}^n & A_{i,j}^n \geq t * \max_{i,j} A_{i,j}^n \\ 0 & \text{else} \end{cases} \tag{12}$$

$$\mathcal{A}_{i,j}^n(t) = \frac{\tilde{\mathcal{A}}_{i,j}^n(t)}{\sum_{i,j} \tilde{\mathcal{A}}_{i,j}^n(t)} \tag{13}$$

Note that $\mathcal{A}_{i,j}^n(0) = A_{i,j}^n$. We run this modified agent on four games — Breakout, Ms. Pacman, Seaquest and Space Invaders — and find that the performance of the agent does not degrade for $t \leq 0.1$. This indicates that the agent is mostly using the information in the regions of high attention and not relying on the softness of the attention to collect information in the tail of the distribution. This gives us confidence that we can rely on the visual inspection of the agent's attention map to indicate what information is being used.

(a) The distribution of attention weights for Ms Pacman.

(b) The distribution of attention weights for Space Invaders

Figure 10: The distribution of attention weights on each head for a Ms Pacman and a Space Invaders frame. The two bar plots show the sum of the weights along the x and y axis (the range of each plot is [0, 1].

### A.7 Attention Weights Distribution

Since the sum that forms the attention answers (Equation 5) runs over all space, the peakiness of the attention weights will have a direct impact on how local the information received by the agent is. Figure 10 shows the distribution of attention weights for a single agent position in Ms Pacman and Space Invaders on all four heads. On both games we observe that some of the heads are highly peaked, while others are more diffuse. This indicates that the agent is able to ask very local queries as well as more general queries. It is worth noting that, since the sum preserves the channel structure, it is possible to avoid washing out information even with a general query by distributing information across different channels.

In section A.6, we ran an agent with a hard cutoff in the attention weights on several games and found that the overall performance on those games is not affected for threshold values $t \leq 0.1$. Table 5 shows the ratio of the score achieved by an agent at $t = 0.1$ to that achieved at $t = 0.0$. We see that the agents are able to achieve broadly similar scores accross a range of games.

| | Experts | | | Multi-level | |
| Level | Feedforward | LSTM | Attention | Feedforward | Attention |
|---|---|---|---|---|---|
| alien | **271.8%** | 0.3% | 206.9% | 26.8% | 27.1% |
| amidar | 50.9% | 2.7% | **1138.9%** | 12.5% | 15.9% |
| assault | 2505.8% | 26.2% | **6571.9%** | 80.3% | 69.5% |
| asterix | 6827.5% | 0.7% | **9922.0%** | 14.2% | 29.5% |
| asteroids | 75.3% | 545.8% | **626.3%** | 1.6% | 2.7% |
| atlantis | **6320.7%** | 6161.6% | 5820.0% | 194.8% | 136.4% |
| bank_heist | 184.0% | **191.8%** | 168.5% | 4.2% | 1.7% |
| battle_zone | 151.9% | **216.2%** | 2.1% | 5.6% | 2.6% |
| beam_rider | **172.3%** | 152.1% | 132.7% | 1.8% | 1.4% |
| berzerk | 39.8% | 353.6% | **1844.3%** | 10.4% | 12.1% |
| bowling | **35.1%** | 1.7% | 9.0% | 3.8% | 3.1% |
| boxing | **832.5%** | 25.2% | 743.6% | 677.1% | 32.5% |
| breakout | **2963.5%** | 2917.4% | 2284.2% | 15.0% | 29.2% |
| centipede | **136.5%** | 12.7% | 108.3% | 43.1% | 35.4% |
| chopper_command | 5885.2% | **8622.1%** | 12.3% | 20.8% | 5.3% |
| crazy_climber | 560.7% | 5.6% | **643.9%** | 374.3% | 398.0% |
| defender | 2835.5% | 3361.2% | **3523.9%** | 98.9% | 76.9% |
| demon_attack | 7406.6% | 7526.0% | **7563.3%** | 47.4% | 112.5% |
| double_dunk | 865.2% | 850.8% | **1934.0%** | 108.4% | 171.6% |
| enduro | **275.0%** | 274.5% | 275.0% | 127.7% | 51.7% |
| fishing_derby | **293.9%** | 8.6% | 280.8% | 132.3% | 10.0% |
| freeway | 0.1% | 0.1% | 0.1% | **75.9%** | 12.9% |
| frostbite | 6.0% | 7.3% | 5.7% | **35.1%** | 4.7% |
| gopher | 4588.1% | 5124.6% | **5280.3%** | 36.4% | 141.6% |
| gravitar | 151.8% | 144.6% | **184.6%** | 3.8% | 3.1% |
| hero | **151.9%** | 6.7% | 121.7% | 43.2% | 22.2% |
| ice_hockey | 241.0% | **302.2%** | 64.1% | 37.7% | 35.6% |
| jamesbond | 845.9% | **5819.2%** | 319.7% | 31.7% | 13.0% |
| kangaroo | **178.9%** | 174.1% | 0.6% | 21.7% | 8.5% |
| krull | 1031.8% | 921.0% | **1309.6%** | 547.4% | 883.3% |
| kung_fu_master | 363.7% | 20.6% | **763.9%** | 73.3% | 118.1% |
| montezuma_revenge | **52.6%** | 0.1% | 0.1% | 0.0% | 0.1% |
| ms_pacman | 195.9% | 6.4% | **442.8%** | 31.6% | 26.4% |
| name_this_game | **482.3%** | 7.5% | 413.1% | 74.0% | 53.9% |
| phoenix | **10705.9%** | 10423.9% | 8560.2% | 47.5% | 63.3% |
| pitfall | **3.4%** | 3.4% | 3.4% | 3.4% | 3.4% |
| pong | **118.1%** | 2.0% | 118.1% | 55.3% | 2.1% |
| private_eye | 0.2% | 0.2% | 1.0% | 0.5% | **2.0%** |
| qbert | 160.6% | 1.2% | **207.7%** | 4.7% | 5.7% |
| riverraid | **118.6%** | -3.3% | 93.4% | 33.8% | 30.9% |
| road_runner | 2441.2% | 2336.6% | **3570.9%** | 409.7% | 284.8% |
| robotank | 625.3% | **700.3%** | 450.3% | 25.6% | 32.1% |
| seaquest | 8.5% | 0.6% | **546.5%** | 1.9% | 1.4% |
| skiing | 63.6% | 63.6% | 8.7% | **63.6%** | 63.4% |
| solaris | 15.7% | **19.1%** | 13.0% | 12.5% | 12.8% |
| space_invaders | 3230.4% | 3412.5% | **3668.0%** | 16.8% | 30.4% |
| star_gunner | 4972.8% | 6707.6% | **6838.6%** | 8.4% | 10.4% |
| surround | 114.2% | 93.0% | **121.9%** | 4.8% | 0.7% |
| tennis | **307.4%** | 153.5% | 0.7% | 49.8% | 45.4% |
| time_pilot | 3511.7% | 16.7% | **5708.4%** | 6.8% | 17.0% |
| tutankham | 169.3% | 19.3% | **187.3%** | 104.1% | 76.9% |
| up_n_down | 4035.0% | 12.3% | **4771.5%** | 347.8% | 59.1% |
| venture | 0.0% | 0.0% | 0.0% | **8.9%** | 3.1% |
| video_pinball | 2853.2% | 139.0% | **3001.8%** | 153.3% | 188.7% |
| wizard_of_wor | **842.5%** | 7.6% | 401.1% | 16.6% | 8.5% |
| yars_revenge | **1100.1%** | 12.7% | 867.0% | 47.8% | 32.2% |
| zaxxon | 472.2% | **521.1%** | 488.6% | 25.5% | 2.8% |

Table 3: The human-normalized score of agents on all ATARI levels.

| level name | Fixed Query Agent | L2 Norm Keys Agent | Top-Down Attention Agent |
|---|---|---|---|
| amidar | 225.7% | 547.5% | **903.6%** |
| asteroids | 88.0% | 126.4% | **541.1%** |
| berzerk | 285.3% | 334.1% | **1153.9%** |
| enduro | 274.8% | 274.5% | **274.7%** |
| ms_pacman | 198.4% | 199.6% | **414.3%** |
| seaquest | **1435.9%** | 49.4% | 28.2% |
| space_invaders | 1798.1% | 2395.2% | **3512.8%** |

Table 4: The scores of the attention agent compared to the two bottom-up experiments described in the text.

| Task | Relative Score |
|---|---|
| Breakout | $88\% \pm 11\%$ |
| Ms. Pacman | $89\% \pm 13\%$ |
| Seaquest | $116\% \pm 29\%$ |
| Space Invaders | $98\% \pm 12\%$ |

Table 5: The score of an agent run with hard attention (equation 13) with $t = 0.1$ as a percentage of the score with $t = 0$. All scores are calculated by running 15 times at each value of $t$. Uncertainties are the statistical uncertainties of the ratio of the mean scores.