[Reviews · NeurIPS 2019]

Reviewer 1



The paper is clear and well-written. Appendix and provided video are also quite helpful to obtain a better understanding of this work. The attention mechanism idea is not quite novel but its usage (considering the modification) in the RL domain is interesting and novel. In my view, Interpretability is the most important aspect of this work. Presentation and the depth of the analyses make this work a good submission and candidate for acceptance.

Reviewer 2



This paper proposes a soft, top-down attention mechanism for reinforcement learning. The main framework is similar to [1], with self-attention replaced by top-down attention generated by an LSTM. The network is adapted to RL scenario where the agent is involved in a sequential decision-making process and no explicit "query" exists. This paper provides a interesting approach to interpretable reinforcement learning. It is clearly written and well-motivated. The authors examined several examples of agent attention and showed interesting behavior of the RL agent. My only concern is that the contribution seems a bit incremental. The recurrent attention structure is not too far from previously proposed . The introduction of spatial basis is interesting but is not completely new either. That being said, I still think the idea and experiments are quite interesting and there are enough new thoughts to benefit future research. Another minor point is that in Table 1 in the appendix it can be seen that sometimes the model with attention performs significantly worse than baseline (e.g. battle zone, chopper command), but to my understanding the agent should always learn to perform at least as well by placing uniform soft attention. Maybe the authors can comment on this? Ref. [1] Ashish Vaswani, Noam Shazeer, Niki Parmar, Jakob Uszkoreit, Llion Jones, Aidan N Gomez, Łukasz Kaiser, and Illia Polosukhin. Attention is all you need. In Advances in Neural Information Processing Systems, pages 5998–6008, 2017.

Reviewer 3



The paper is well-written and clear; the architecture is described in detail through a diagram (Figure 1 on page 2), with the math in section 2 expanding on the key components of the attention mechanism. High-level details for the RL training setup, implemented baselines, and condensed results are provided in the body of the paper. Detailed learning curves for each of the compared approaches are presented in the appendix (which is appropriate, given that the task-specific learning performance is secondary to the analysis of the attention mechanism). The analysis section is thorough, and I specifically appreciated the section at the end comparing the learned attention mechanism to prior work on saliency maps. Model/Architecture Notes: While the proposed model is a straightforward extension of query-key-value attention to tasks in RL, there are two interesting architectural features: First, “queries” for their attention mechanism can be decomposed into features that act on content (which the paper refers to as the “what”), and features that act on spatial location (which the paper refers to as the “where”). Second, the attention mechanism is described as being “top-down” rather than “bottom-up”; the queries used are computed from the “query network” which is *not dependent on the input image*, and instead *conditioned on solely the prior state*, capturing macro-level information about the task, rather than local/spatial information about a given frame. Analysis Notes: I found the analysis to be insightful and well-conducted. I specifically found the sub-section 4.2 (“Reaction to novel states”) extremely convincing, showing how the learned agent behaves when injecting a known object (“fish”) at an unexpected time and location. Furthermore, I found sub-section 4.5 (“Trip wires”) interesting - seems to support the claim that attention does allow us to better interpret agent behavior. --- Key Questions/Concerns: The biggest question I have is in regards to the discussion of the role of “top-down” influence in the attention mechanism in sub-section 4.4. Specifically, it seems that there are two key features present in the “top-down” mechanism: 1) Focus on Macro/Task-Level Features: The mechanism is able to focus on high-level features of the task, instead of local, image/frame-specific features. This seems to be the feature that the paper is trying to highlight in its discussion of differences between “top-down” and “bottom-up” influence. 2) Time-Dependent Query Generation: The mechanism is able to generate different queries at different points in time. Unfortunately, it isn’t clear from the current experiments which of these features is responsible from the gains in performance observed over the two presented baselines: 1) learning a fixed query tensor, and 2) computing attention weights directly from the keys. I think that this distinction about top-down/bottom-up attention is a crucial point in this paper, and I’d like to believe that it is because of point (1) above. To make this clear in the experiments, however, I suggest implementing a bottom-up baseline that allows for time-dependent attention masks (ablate out (2) above). One way to do this is to learned a fixed attention mask for each time point t in the episode (for example, compute attention mask as a function of a position encoding that captures timestep). The second question that I have regards the failure modes of the attention-augmented agent. In the appendix, it seems there are several games on which the attention-augmented agent performs notably worse than the Feed-Forward and LSTM baselines (e.g. “chopper_command” and “jamesbond“). I wonder if the learned attention mechanism provides any indication as to why the agent is failing at these tasks - and if so, I think such a discussion would be very useful in this paper. If the learned mechanism doesn’t provide any immediately apparent explanation, some possible explanations as to why the baseline agents perform better would be appreciated (similar to the notes made about why bottom-up attention is better for ”Seaquest“ in sub-section 4.4). Nitpicky Things: I couldn’t get the videos on the supplemental site to play, unfortunately. Despite viewing the paper on a computer monitor, I found several of the visualized attention images hard to read... perhaps fewer images per row/bigger images would help? May also be nice to have more examples in the appendix (where you can increase size of figures significantly). I found this sentence in the second paragraph on page 4 hard to parse: “... This enables our system to be both state/context dependent and input dependent.” A possible rewording to make the (presumed) point a little more clear is: “... This enables our system to be both state/context dependent (by informing the queries from this top-down source) as well as input dependent (through the ”vision core“)”.

[Author Response · NeurIPS 2019]

We thank the reviewers for their helpful and constructive comments and we are happy they found the paper interesting.

**Reviewer 1**    Reviewer 1 suggested we investigate the influence of top-down vs. bottom up attention — we agree this is one of the more interesting aspects of the model. In Section 4.4 of the paper we provide some quantitative studies of the advantage of the top-down attention over the bottom up, but we treated it more as an ablation study to demonstrate that the top-down queries were playing an important role . This also influenced the style of bottom-up attention we chose: the one in the paper keeps most of the model the same and isolates actual top-down influence. We agree that a full analysis of the strategies that a bottom-up agent learns, including a detailed comparison with a top-down agent, would be very interesting, but length constraints prevented further analysis here.

**Reviewer 2**    Reviewer 2 was concerned about novelty of the model. We agree that similar models have been proposed in the past and we note this in the paper. We feel that the novel contribution of the paper is the application of an attention model to RL and the interpretability of the rich behaviors and strategies that emerge from this.

Another concern was about what happens when the agent doesn't learn well (such as in chopper command and battle zone) and why uniform attention doesn't help in that case. Uniform attention, like having an average pooling layer over the entire conv-net output, causes the information to be blurred across the whole image due to the spatial reduce sum (Equation 5. in the paper) — this will prevent the agent from extracting local information from images and often fail at the task. We also think that some long-term credit assignment issues may be exacerbated by using this attention, since the agent has to be already be attending correctly to even assign the reward to the right place. This would be an interesting avenue for further research.

**Reviewer 3**    Like Reviewer 1, Reviewer 3 was also interested in the effects of top-down influence and raises an interesting separation of the two salient features of the top-down mechanism - one is that it has to do with the macro/task level features, and the other is the time dependency. We argue that the it is indeed the macro level influence (as R3 suggests) and that the experiments in Section 4.4 show that. The reason time dependence is not the key factor here is that both non top-down baselines can be time-dependent - the ConvLSTM, which has a state, can carry information forward in time. What it can not do is to carry that information globally, but just locally through convolutions. So whether it's creating key/values for the fixed query as in the first baseline, or explicit attention maps, both **can** be time dependent, but can't be macro/task dependent. While this is different than the test that R3 suggests, we believe one can draw similar conclusions from it. The ConvLSTM could learn to sequentially promote different features to the learned, fixed query set which would have a similar effect to learning sequential masks.

We apologize that the videos were not available, we're not sure why (R1 was able to view them), but we suggest trying again now.

We will clarify the text to make the last point R3 raises clearer.

[Meta-Review · NeurIPS 2019]

The reviewers appreciated the analysis of the learned attention from the model, and recommend accepting. I would ask that the authors add some of the experiments suggested by R4 in their "key questions" section. If there is not enough space, please add these to the supplemental.